# Pharmacological Inhibition of Membrane Signaling Mechanisms Reduces the Invasiveness of U87-MG and U251-MG Glioblastoma Cells In Vitro

**DOI:** 10.3390/cancers15041027

**Published:** 2023-02-06

**Authors:** Alanah Varricchio, Sidra Khan, Zoe K. Price, Rohan A. Davis, Sunita A. Ramesh, Andrea J. Yool

**Affiliations:** 1Discipline of Physiology, School of Biomedicine, University of Adelaide, Adelaide, SA 5005, Australia; 2Obstetrics and Gynaecology, Adelaide Medical School, University of Adelaide, Adelaide, SA 5005, Australia; 3Griffith Institute for Drug Discovery, School of Environment and Science, Griffith University, Nathan, QLD 4111, Australia; 4Biological Sciences, College of Science and Engineering, Flinders University, Bedford Park, SA 5042, Australia; 5Discipline of Physiology, School of Biomedicine and the Institute for Photonics and Advanced Sensing, University of Adelaide, Adelaide, SA 5005, Australia

**Keywords:** ion channel, glutamate receptor, aquaporin, natural products, caelestine C, xanthurenic acid, brain cancer

## Abstract

**Simple Summary:**

Glioblastoma is a brain tumor that is among the deadliest of human cancers. The invasive spreading of the tumor cells allows subpopulations to escape chemo and radiation treatments and infiltrate other areas of the brain where they continue to grow and cause relapses. To achieve their high motility, glioblastoma cells appear to harness membrane signaling proteins including aquaporin water channels, glutamate receptors, and ion channels. Work here tested selected combinations of agents that selectively targeted sets of signaling proteins known to be enriched in glioblastomas and showed that blocking both glutamate receptor and aquaporin-1 channels is an attractive means for limiting tumor cell motility without causing cell toxicity that might negatively impact normal brain cell function. Using combinations of agents at threshold doses could enable focusing of the pharmacological effects on cancer cells specifically, minimizing off-target effects.

**Abstract:**

Impairing the motility of glioblastoma multiforme (GBM) cells is a compelling goal for new approaches to manage this highly invasive and rapidly lethal human brain cancer. Work here characterized an array of pharmacological inhibitors of membrane ion and water channels, alone and in combination, as tools for restraining glioblastoma spread in human GBM cell lines U87-MG and U251-MG. Aquaporins, AMPA glutamate receptors, and ion channel classes (shown to be upregulated in human GBM at the transcript level and linked to mechanisms of motility in other cell types) were selected as pharmacological targets for analyses. Effective compounds reduced the transwell invasiveness of U87-MG and U251-MG glioblastoma cells by 20–80% as compared with controls, without cytotoxicity. The compounds and doses used were: AqB013 (14 μM); nifedipine (25 µM); amiloride (10 µM); apamin (10 µM); 4-aminopyridine (250 µM); and CNQX (6-cyano-7-nitroquinoxaline-2,3-dione; 30 µM). Invasiveness was quantified in vitro across transwell filter chambers layered with extracellular matrix. Co-application of each of the ion channel agents with the water channel inhibitor AqB013 augmented the inhibition of invasion (20 to 50% greater than either agent alone). The motility impairment achieved by co-application of pharmacological agents differed between the GBM proneural-like subtype U87-MG and classical-like subtype U251-MG, showing patterns consistent with relative levels of target channel expression (Human Protein Atlas database). In addition, two compounds, xanthurenic acid and caelestine C (from the Davis Open Access Natural Product-based Library, Griffith University QLD), were discovered to block invasion at micromolar doses in both GBM lines (IC_50_ values from 0.03 to 1 µM), without cytotoxicity, as measured by full mitochondrial activity under conditions matching those in transwell assays and by normal growth in spheroid assays. Mechanisms of action of these agents based on published work are likely to involve modulation of glutamatergic receptor signaling. Treating glioblastoma by the concurrent inhibition of multiple channel targets could be a powerful approach for slowing invasive cell spread without cytotoxic side effects, potentially enhancing the effectiveness of clinical interventions focused on eradicating primary tumors.

## 1. Introduction

Glioblastoma (also known as glioblastoma multiforme; GBM) is classed as a Grade IV glioma, among the deadliest of human cancers, and is the most frequently occurring type of brain tumor [1,2]. The rapid growth of glioblastoma and the ensuing pressure on the brain can manifest as symptoms including chronic headaches, seizures, and impaired motor and cognitive functions depending on the brain regions involved [3]. The median survival time for glioblastoma patients is poor, typically 12–14 months following diagnosis; unfortunately, therapeutic benefits of available therapies have remained limited despite decades of dedicated effort to improve treatments [4]. Current strategies combine radiation, chemotherapy, and surgical resection where possible; however, at best, they enable temporary reductions in symptoms and small increases in patient longevity [2]. Diffuse infiltration of glioma cells into healthy brain parenchyma remains the major challenge for treatment strategies aimed at eradicating primary tumors and is further complicated by the multiforme (heterogeneous) nature of GBM tumors. Complex arrays of gene deletions, amplifications, and mutations in GBM subtypes result in varying levels of resistance to therapies and enhancements of signaling pathways linked to proliferation and motility [5]. Heterogeneity increases the risk that subpopulations of glioblastoma cells will escape treatments, continue to propagate, and cause relapses. Investigating pharmacological agents that target GBM cell motility has become a subject of clinical trials going beyond traditional chemotherapeutic methods. Cilengitide, an integrin inhibitor that disrupts endothelial cell migration and adhesion [6], has proven promising in phase II clinical trials for recurrent glioblastoma patients, who had a 6-month progression-free survival rate of 15% and a median overall survival of 9.9 months with no reproducible toxicities observed after receiving cilengitide twice weekly [7]. 

In GBM invasion, cancerous cells disseminate from a primary tumor mass into neighboring brain regions in a multi-step process that depends on cell motility, cell-cell and cell-matrix adhesions, cytoskeletal rearrangement, regulation of proteases, and extracellular-matrix (ECM) degrading molecules [8,9,10,11,12,13,14,15,16,17,18,19,20,21] to allow penetration through brain interstitial spaces [22]. Signal transduction cascades enabling motility have been suggested to harness membrane signaling proteins including aquaporins (AQPs), glutamate receptors, and ion channels; specific classes have been highlighted by patterns of increased transcription in biopsied human GBM samples [23], and prior links to enhanced motility in other cell types [24,25].In the aquaporin family, AQP1 is an intriguing target for anti-motility therapeutics development, with published work showing AQP1 channels clustered on the leading edges of glioblastoma cells provide accelerated motility in disease progression [26], and AQP1 channels localized at leading edges of neural crest cells enable the rapid migration essential for normal nervous system development [27]. Quantitative analyses of the Human Protein Atlas transcriptomic database (https://www.proteinatlas.org; accessed on 12 August 2022) showed transcript levels are enriched for AQP1 and specific classes of ion channels in glioblastoma biopsy samples (*n* = 153), with ‘enrichment’ defined as transcript levels at least fourfold higher than those in other tissues (fragments per kilobase of transcript per million fragments mapped) [23]. Enriched expression levels were observed for the AMPA glutamate receptor GluR2, voltage-gated Ca^2+^ channel Ca_V_1.1, voltage-gated K^+^ channel K_V_3.1, and Ca-activated K^+^ channels K_Ca_2.3 and K_Ca_2.2 [23,24,28].

We tested the hypothesis that glioblastoma cell invasiveness can be controlled by the combined pharmacological inhibition of AQP1 and ion channels known to be upregulated in GBM. Co-administration of channel antagonists at low doses could exploit patterns of channel expression which are distinctive to GBM cancer subtypes and minimize toxic effects on healthy brain cells. Non-toxic pharmacological treatments that overcome the aggressive invasiveness of GBM cells could be used to restrain invasion, holding GBM cells in place while primary tumor eradication treatments were applied. 

The use of natural compounds to treat human diseases dates back to ancient civilizations [29]. Opium, digitalis, and aspirin are examples of major pharmaceutical products in the modern armamentarium; more remain to be discovered [30]. More recently, venom peptides containing bioactive compounds also have inspired interest for therapeutic strategies [31,32]. Chlorotoxin, from the scorpion *Leiurus quinquestriatus*, binds the chloride channel ClC-3 with high affinity, and holds promise as a cell-specific vehicle for targeting treatments selectively to glioblastoma tumor cells [21,33,34]. Apamin from the honeybee *Apis mellifera* blocks low conductance Ca^2+^-dependent K^+^ channels [35], and was found to exert anti-tumor effects in human colon cancer cells by inhibiting the NF-κB signaling pathway, which regulates tumor angiogenesis and invasiveness [36,37]. Melittin, another component of bee venom, directly inhibited the invasive and migratory ability of metastatic breast cancer cells by suppressing metalloproteinase-9 expression [38]. 

We report here that a selected set of pharmacological agents and natural compounds inhibit invasion in glioblastoma cell lines and offer promise for potential intervention strategies in these highly malignant tumors. The list of candidate agents included two related compounds (xanthurenic acid and caelestine C), shown in Figure 1, that are thought to be modulators of glutamatergic neurotransmitter activity in synapses. Xanthurenic acid (XA) is a tryptophan metabolite natively present in the CNS at micromolar concentrations and demonstrated to promote activation of metabotropic glutamate receptors (subtypes mGluR 2, 3) [39,40,41], and to inhibit vesicular glutamate transport [42]. Caelestine C is a brominated quinoline carboxylic acid from the Australian ascidian *Aplidium caelestis* [43], not previously evaluated for biological signaling activity. Chemical analogs of caelestines with minor differences in benzene ring substituents have been found to differentially antagonize NMDA (N-methyl-D-aspartate) and non-NMDA glutamate receptors [43,44] (see Discussion for details). 

A general concern with ion channels and receptors as candidate GBM targets is that these signaling pathways also are normal mechanisms of function in neurons and glia. We propose that using combinations of agents at threshold doses could enable focusing of the pharmacological effects on brain cancer cells, taking advantage of GBM subtype ‘fingerprints’ (i.e., the class-specific patterns of upregulated ion and water channels in the cancer cells), a step towards the goal of minimizing off-target effects.

## 2. Materials and Methods

### 2.1. Cell Culture

Human glioblastoma cell lines U87-MG and U251-MG (American Type Culture Collection, Manassas, VA, USA) and the astrocytoma cell line 1321N1 (Merck, formerly Sigma-Aldrich; Bayswater VIC Australia) were cultured in Dulbecco Modified Eagle Medium (DMEM; Gibco) containing 10% fetal bovine serum (FBS), 1% GlutaMAX, and 100 units/mL each of penicillin and streptomycin. Cell culture media and reagents were from ThermoFisher Scientific (Melbourne, VIC, Australia) unless otherwise indicated. All cell cultures were grown at 37 °C in a humidified 5% CO_2_ incubator. To minimize phenotypic variation, cell cultures were maintained < 1 month after re-animation and passage numbers used for experimental work ranged from 9 to 12 [45]. Properties of the 1321N1 cell line are comparable to those of native astrocytes and include the transport of dopamine, serotonin, norepinephrine, and histamine [46].

### 2.2. Pharmacological Agents

Nifedipine, amiloride, apamin, 4-AP (4-aminopyridine), and CNQX (6-cyano-7-nitroquinoxaline-2,3-dione) were purchased from Merck. AqB013 [47] was synthesized by Dr G. Flynn (Spacefill Enterprises LLC, Bozeman, MT, USA) and repurified by Dr. N. Proschogo (University of Sydney, NSW, Australia). Xanthurenic acid and caelestine C from the Davis Open Access Natural Product-based Library were provided by R. Davis (Griffith University, QLD Australia). Stock solutions (1000×) of AqB013 (14 mM), nifedipine (25 mM), amiloride (10 mM), CNQX (30 mM), xanthurenic acid (1 mM), and caelestine C (1 mM) were prepared in DMSO. DMSO alone at 0.1% served as the vehicle control for DMSO-solubilized compounds. Stock solutions (1000×) of apamin (10 mM) and 4-aminopyridine (250 mM) were prepared in water; for these, media without DMSO served as the control comparison. The tested concentrations of AqB013, nifedipine, amiloride, CNQX, apamin, and 4-aminopyridine were selected based on parameters established in the literature [48,49,50,51,52]. For the novel compounds, xanthurenic acid and caelestine C, dose-response experiments were conducted to determine the concentrations to be used for the cellular motility assays.

For testing in transwell invasion assays, all pharmacological agents were diluted at 1 µL/mL in final media. For testing in spheroid spreading assays, all pharmacological agents were diluted at 2 µL/mL in final media and DMSO alone at 0.2% served as the vehicle control for DMSO-solubilized compounds.

### 2.3. Transwell Invasion Assays

Invasion assays were performed, as previously described [53], in 24-well plates with transwell filter inserts (6.5 mm, 8 µM pores; Corning transwell polycarbonate; Merck) layered with extracellular matrix gel (40 μL/well, Matrigel; Merck, diluted 1/30 and 1/40 for U87-MG and U251-MG respectively). Briefly, U251-MG and U87-MG cultures were grown to approximately 40% confluency before starvation in DMEM containing reduced (2%) FBS. After 24 h, cells were detached (with 1% trypsin and 0.5% EDTA in PBS) and resuspended in FBS-free DMEM. Cells were seeded on transwell filter inserts (total 150 μl of cell suspension per well, including 50 µL of FBS-free rehydration medium) at 5 × 10^4^ cells/mL per well in the presence of the appropriate concentration of drug-treated or vehicle control medium. The chemoattractant gradient in each well was created using 10% FBS in the basal chamber solution only. After incubation (37 °C in 5% CO_2_) for 4 h (U87-MG) or 4.5 h (U251-MG), non-invasive cells were removed from upper (cis) surfaces of filters. Invasive cells on bottom (trans) surfaces were stained with crystal violet (0.2% *w*/*v*, Merck) and live imaged with EOS Utility 3 (Canon, USA) on an inverted microscope (ULWCD 0.30, Olympus Corp., Tokyo, Japan). Percent invasion was calculated as the average number of invaded cells per well (number on trans filter sides) compiled for three randomly selected fields (20× objective) and standardized to the mean value of the matched vehicle or untreated control. The acquisition of invasive morphologies was quantified by measuring the elongation index of cells that had traversed the transwell filter. The elongation index was calculated as the ratio of the major and minor axes for at least five cells in three independent fields of view for each treatment type. Independent experiments were repeated at least twice, with three to twelve replicates per treatment group.

### 2.4. Spheroid Spreading Assays 

Cell suspensions of U87-MG and U251-MG grown in DMEM (10% FBS) were plated at 100 µL volumes (2.5 × 10^3^ cells/well) in 96-well round bottom Costar plates (Corning Inc., Corning, NY, USA) coated prior to use with 30 mg/mL poly(2-hydroxyethyl methacrylate) (polyHEMA; Merck), using methods described previously [54]. Briefly, plates were then centrifuged (200× *g*; 5 min) to sediment cells, and incubated 48 h (37 °C, 5% CO_2_) to promote aggregation in spheroids. Plates were then cooled on ice (15 mins), 50 µL of Matrigel was added to each well, and plates were incubated overnight. The culture medium was exchanged with fresh medium containing test compound or control treatments and the initial set of images was taken (0 h). Time-lapse images were taken with 10× magnification at 24, 48, 72, and 96 h (Olympus Phase Contrast ULWCD microscope; Notting Hill, VIC Australia) to allow quantification of spheroid growth and particle dispersal (invasive spreading) rates using ImageJ software (version 1.53t (August 2022 update), first published in 2012, https://doi.org/10.1038/nmeth.2019 (accessed on 10 January 2023), U.S. National Institutes of Health, Bethesda MD, USA; https://imagej.nih.gov/ij/download.html (accessed on 10 January 2023)). To analyze spheroid growth, spheroid perimeters were drawn with the ImageJ freehand tool. Total perimeter distance was plotted as a function of treatment duration for each spheroid, and data were analyzed by linear regression to generate slope values reflecting rates of perimeter increase. To measure rates of dispersal, 8-bit greyscale images were compiled into time-series stacks for each spheroid, converted to binary, and threshold settings were adjusted empirically to optimize clarity and contrast for each complete image stack before semi-automated particle detection. Software parameters for size (square pixels) were set at 0–Infinity, and for circularity were set at 0.00–1.00. Numbers of particles that emanated beyond the core spheroid perimeter were quantified by software detection at each time point, plotted as a function of time, and analyzed by linear regression to generate slope values corresponding to rates of particle dispersal. Rates for treatment groups were standardized to mean rates for matching vehicle or untreated control groups. Two independent experiments were conducted for each treatment, with three to five replicates in each.

### 2.5. Cytotoxicity Assays

Cells were seeded in 96-well plates in DMEM media with 10% FBS and incubated overnight. After application of media containing test compounds or vehicle controls, cultures were incubated 4 h (for U87-MG) or 4.5 h (for U251-MG and astrocytes), matching conditions used for transwell invasion assays. Cell viability was measured using the Alamar Blue assay [55] following manufacturer’s instructions (ThermoFisher Scientific). Briefly, cells in DMEM (10% FBS) were treated 90 min with 10% Alamar Blue solution; fluorescence was measured using a FLUOstar Optima microplate reader (BMG LabTech, Mornington, VIC, Australia). A reference sample with DMEM medium only (no cells) was included to establish the background signal level. 

### 2.6. Statistical Analyses

Statistical analyses were performed with GraphPad Prism software (version 9.0.0 (121), 22 October 2020, San Diego, CA, USA) using one- or two-way ANOVA with post-hoc tests as specified in Figure legends. Statistically significant results are represented by asterisks as **** *p* < 0.0001, *** *p* < 0.001, ** *p* < 0.01, * *p* < 0.05, and ns (not significant). Boxplots show distributions of the central 50% of data values (in boxes), the full range of data (error bars), and median values (horizontal bars). Bar histograms show the mean ± standard deviation. *n*-values are above the x-axes.

## 3. Results

The effects of pharmacological compounds on glioblastoma invasion were assessed individually and in combination with the AQP1 water channel inhibitor AqB013 in assays using transwell filters with extracellular matrix. In U87-MG cells, all treatments elicited at least 40% block of invasion (Figure 2A) as compared to matched controls (‘DMSO’ for AqB013, nifedipine, amiloride, CNQX, XA, and caelestine C, shown with black asterisks; or ‘untreated’ for apamin and 4-AP, shown with blue asterisks). As compared with AqB013 alone, invasion was more strongly impaired when Ca^2+^, Na^+^ or K^+^ channel blocking agents (nifedipine, amiloride, apamin, and 4-AP) were co-administered (red asterisks). As compared with each ion channel blocker alone, co-application of the AqB013 augmented the block of invasion with nifedipine, amiloride, apamin, and 4-AP (green asterisks). When combined with CNQX, XA, or caelestine C, a trend towards additional block by AqB013 was observed though did not reach statistical significance as compared to individual agents alone (in green).

In U251-MG cells, all pharmacological inhibitors (except 4-AP) applied singly blocked invasion by 20 to 60% as compared with matched controls, and 4-AP showed a strong inhibitory trend (Figure 3A). Co-application of AqB013 augmented the invasion-blocking effects of apamin, 4-AP, XA and caelestine C. Representative images of transwell-migrated U87-MG and U251-MG cells (Figure 2B and Figure 3B respectively) are shown for control, caelestine C, AqB013, and combined treatments as indicated.

An elongation index was quantified as the ratio of the maximum divided by the minimum cell diameters for cells that had traversed the transwell filter in the different treatment groups (Appendix A). The results showed that the morphology of highly invasive cells was associated with a higher elongation index, indicating the adoption of an elongated, spindle-like shape, as seen in untreated and DMSO controls that passed through the transwell filter; conversely all treatments which reduced motility impaired morphological elongation.

None of the pharmacological agents induced cytotoxicity in glioblastoma cells or astrocytes, as assessed by Alamar Blue assays (Figure 4) for conditions matching those used in transwell assays, confirming that impaired invasion did not result indirectly from a loss of cell viability. 

Anti-invasive effects of channel-blocking compounds identified in the transwell assays (see Figure 2 and Figure 3 above) were evaluated in the spheroid model for both GBM cell lines, for results measured as numbers of cell clusters (particles) that detached from the spheroid mass and migrated into the surrounding extracellular matrix environment. Spheroid perimeter was the metric used to determine the effects of the individual and combined drug treatments on growth and survival. 

Results for cell detachment and migration away from U87-MG spheroids are summarized in Figure 5. Rates of particle dispersal were significantly reduced by all single agents except XA and caelestine C as compared to matched controls (Figure 5A). None of the agents when combined with AqB013 were better than the single agents alone in reducing invasiveness, except XA and caelestine C. The pharmacological treatments did not compromise spheroid growth (Figure 5B), indicating that reduced migration was not indirect consequence of reduced cell viability. Numbers of particles disseminated from individual spheroids were plotted as a function of treatment time (Figure 5C) and well fit by linear regression (all but five R values ≥ 0.8). The average slope values (mean ± SD) for rates of particle dispersal were 14.8 ± 2.1 for vehicle control; 12.8 ± 1.9 for caelestine C (*n* = 4 per treatment); 6.83 ± 2.4 for AqB013; and 6.04 ± 1.2 for the combined treatment (*n* = 3 per treatment). Representative images illustrating numbers of cell cluster particles that invaded the surrounding ECM by 96 h are shown in Figure 5D,E for single treatments, and in Figure 5F for combined, as indicated. Inhibitors blocked U87-MG invasiveness in the spheroid model without impairing viability. 

Results for cell cluster particle dispersal from U251-MG spheroids in different pharmacological treatments are shown in Figure 6. U251-MG spheroids showed reduced rates of particle dissemination in all treatments with single agents (Figure 6A), including those that were not effective alone for U87-MG (XA and caelestine C; see Figure 5A). Co-treatment with AqB013 did not augment the blocking effect of any agents, whether compared to AqB013 alone, suggesting maximal inhibitory effects had been reached in the individual treatments. No impairment of spheroid growth was observed in control or treatment conditions (Figure 6B), indicating cell viability was not compromised. The numbers of particles dispersed from U251-MG spheroids increased linearly over 96 h (Figure 6C); rates were reduced by either the single or by combined treatments as compared to controls. The average slope values (mean ± SD) for rates of particle dispersal were 10.2 ± 1.6 for vehicle control; 6.19 ± y = 1.1 for caelestine C (*n* = 4 per treatment); 7.20 ± y = 1.5 for AqB013; and 5.67 ± y = 1.06 for the combined treatment (*n* = 3 per treatment). Representative images at 96 h are shown in Figure 6D–F. Images of U87-MG and U251-MG spheroids at 0 h and 96 h post-treatment with all other pharmacological agents (alone and in combination with AqB013) are shown in Appendix A.

The inhibitory effects of XA and caelestine C on U87-MG and U251-MG transwell invasion though an ECM barrier layer were dose-dependent, as shown in Figure 7. Numbers of cells that traversed the filters in response to a standard chemoattractant gradient (fetal bovine serum in the basal chamber) were quantified by assessors blinded to treatment groups. Equivalent numbers of cells were loaded per well. Numbers of invaded cells were standardized as a percentage of the mean number of invaded cells in vehicle control. Estimated IC_50_ values (shown in figure keys) indicated potent anti-invasive activities at 0.2 to 0.3 µM concentrations for U87-MG (Figure 7A), and at 0.2 to 1 µM for U251-MG (Figure 7B). 

## 4. Discussion

Outcomes of this work support the idea that channel inhibitors might be repurposed to control motility in GBM cells; combinations of ion channel pharmacological agents with an AQP1 water pore blocker created strong anti-invasive treatments for GBM cell lines. These effects were accompanied by minimal cytotoxic effects in astrocytes, which is progress towards the objective to limit off-target effects that affect normal neuronal and glial functions. The new portfolio of compounds identified here is a promising roster of candidates for brain cancer therapy that merits expanded testing and confirmation in ex vivo and in vivo preparations.

Improved options are needed for treating clinically challenging diseases such as glioblastoma. One of the major outcomes here is the demonstration of augmented block of GBM cell invasiveness using combinations of pharmacological agents. Multiple pathways are required in the motility process, consistent with the finding that combined treatments in general work better than single agents as therapeutic strategies. A second outcome is the demonstration that xanthurenic acid and caelestine C block glioblastoma invasion at micromolar levels without cytotoxic effects, as shown for both the U87-MG and U251-MG GBM cell lines. This block of cell invasion was dramatic and offers an exciting incentive for further testing in ex vivo and in vivo models. 

The only biological activity reported previously for caelestine C involved testing this natural product against three mammalian cell lines (MCF-7, NFF, and MM96L) and a panel of microbial strains [43]. Caelestine C displayed no antimicrobial activity at concentrations up to 10 mM, and partial inhibition (<59%) of growth in the mammalian cell lines at 100 µM, consistent with results here showing minimal cytotoxicity at 100-fold lower doses [43]. The caelestine C and xanthurenic acid compounds tested here are closely related (as illustrated in Figure 1), both classified as quinolines. This particular structural class includes many agents that act as glutamate receptor subtype-selective agonists and antagonists, which differ subtly in terms of the positions and types of meta substituents incorporated in the benzene ring of quinoline. Caelestine C, originally isolated by Davis and colleagues [43], has not previously been characterized for biological signaling activity. However, an agent almost identical to caelestine C that lacks the 8-methoxy group was found in a broad screen by Leeson and colleagues to act as a selective antagonist of non-NMDA glutamate receptors, preferentially inhibiting the AMPA/kainate subtype with an IC_50_ value of 28 µM at quisqualate-activated receptors and 30 µM at kainate-activated receptors, in contrast to low potency at the NMDA receptor with an IC_50_ value > 100 µM [56]. A related xanthurenic acid analog (which has 7-Br rather than 6-Br and lacks the 8-methoxy group of caelestine C), in contrast, preferentially blocks the NMDA glutamate receptor by binding at the glycine modulatory site [56]. Showing a similar mechanism of action, kynurenic acid (which lacks any benzene ring substitution) suppresses hyperexcitability in the CNS by blocking the NMDA receptor glycine site [44], as do the agents 5,7-dichlorokynurenic acid [57] and 7-chlorokynurenic acid [58]. An 8-hydroxy group in the benzene ring of XA (not present in kynurenic acid) is suggested to account for the ability of XA to specifically block glutamate uptake into synaptic vesicles [59]. Conversely a 6-hydroxy group in 6-hydroxykynurenic acid (not present in kynurenic acid) changes its biological action to a competitive inhibitor, which is better at blocking AMPA (IC_50_ 22 µM) than NMDA (IC_50_ 59 µM) receptors [44]. The AMPA/kainate antagonist activity proposed for caelestine C is consistent with our observed anti-invasive effects of another AMPA/kainate receptor antagonist, CNQX, as described in Figure 2, 3, and 5 above. 

Xanthurenic acid has been proposed as a putative neurotransmitter in the CNS, based on observations that it is stored in synaptic vesicles and shows Ca^2+^-dependent release in response to depolarization [60]. XA acts on Group II metabotropic glutamate receptor subtypes 2 and 3, at which it is thought to serve as an allosteric agonist with EC_50_ values in the nM range [61]. Activation of the mGluR2 and its three subtypes in the CNS are associated with inhibitory responses, which negatively regulate the excitatory actions of other CNS neurotransmitter pathways such as serotonin and dopamine [61,62]). This potent effect in suppressing excitatory neurotransmitter pathways is consistent with results here if XA directly or indirectly downregulates glutamate-activated pathways in GBM. Calcium-permeable AMPA/kainate glutamate receptors are implicated in GBM survival and invasion [63], and mechanisms that counteract this activation would be expected to oppose GBM progression. Separate actions of XA as a competitive blocker of the vesicular glutamate transporter require nearly 200-fold higher concentrations than those tested here, making this less likely to be the mechanism for anti-invasive effects of XA. 

Data here suggest that non-NMDA receptors and AQP1 channels might serve complementary roles in enhancing invasive cancer activity, since blocking both was generally more effective than blocking either alone. Given the essential biological roles that AQPs mediate in almost all types of cells, it is not surprising that aberrant AQP functionality has been implicated in a variety of pathological conditions including cerebral edema, various types of major organ failure, metabolic disease, and cancer progression, to list a few [64,65,66,67,68,69,70,71,72,73,74,75,76,77]. AQP1 channels promote invasion of glioblastoma tumor cells through the ECM and into surrounding tissues [23]. AQP4, naturally abundant in astroglia throughout the CNS [78], is not a prognostic indicator for GBM [24]. Previous studies demonstrated that pharmacological inhibition of AQP1 water and ion channels was an effective tool for limiting colon cancer cell motility measured in both wound closure assays and transwell invasion tests [79].

Differences reported here in the sensitivities of U87-MG and U251-MG cells to ion and water channel inhibitors of invasiveness could be expected to reflect the patterns of expression of the corresponding channels and receptors. This prediction was tested by comparing the pharmacological sensitivities with the reported transcript levels for targets of interest in human GBM biopsies (data available from the GBM BioDiscovery Portal database; https://gbm-biodp.nci.nih.gov, last accessed on 4 April 2022), as summarized in Figure 8, quantified as FPKM (Fragments Per Kilobase of transcript per Million mapped reads) and presented as Z-scores calculated for each patient (distance from the population mean, measured in standard deviations). For the comparison, we needed to know the subtypes of GBM that are represented by the cell lines; however, neither U87-MG nor U251-MG have previously been classified with respect to recognized GBM subtypes [80]. Thus, we analyzed their expression profiles (data available from the Broad Institute Cancer Cell Line Encyclopedia, CCLE; https://sites.broadinstitute.org/ccle, last accessed on 11 September 2022) for the presence of diagnostic molecular markers [81,82]. EGFR amplification in U251-MG was consistent with a classical subtype [83]. Upregulation of cyclin-dependent kinase CKD4 and mutation of transcriptional regulator ATRX [83] in U87-MG were consistent with a proneural subtype. Thus, the classifications of U251-MG and U87-MG used for this analysis are classical and proneural subtypes, respectively.

Expression of *GRIA2*, *CACNA1C*, and *KCNC1* is higher in the proneural than the classical subtype (Figure 8). The inhibitors that gave the greatest block of inhibition alone were nifedipine and 4AP in the U87-MG cells. These agents also inhibited invasion in U251-MG though slightly less potently, fitting the lower levels of expression in the classical subtype. Furthermore, also fitting the pattern, CNQX inhibited invasion in both cell lines, but was better in U87MG than U251-MG. Amiloride blocked invasion to similar extents in U87-MG and U251-MG, reflecting the similar expression levels of *SCNN1A*, which encodes amiloride-sensitive ENaC channels, observed in the proneural and classical subtypes. The additive inhibition of invasion by CNQX combined with AqB013 was specific to U87-MG and might reflect the higher transcript levels of the AMPA/kainate receptor GluR-2 (*GRIA2*), one of the targets of CNQX, in the proneural cohort. In U251-MG, only the combinations of apamin, 4-aminopyridine, or nifedipine with AqB013 produced additive block of invasion, an observation that could correlate with lower transcript levels of *GRIA2*, *CACNA1C*, and *KCNC1* in the classical subtype. Inhibitory effects of nifedipine suggest a role in invasion for high-voltage-activated classes of Ca^2+^ channels, which previously have been implicated in cellular motility pathways [40,41,84]. Transcript levels for the nifedipine-sensitive classes Ca_V_1.2 and Ca_V_1.3 are higher in proneural than classical subtypes, but blockers produced similar levels of inhibition of invasion in the two GBM cell lines. Similarly, the voltage-gated K^+^ channel K_V_3.1 (*KCNC1*) has a higher transcript level in proneural; however, 4-aminopyridine combined with AqB013 elicited an additive block of invasion in both U87-MG and U251-MG. Results with the Ca^2+^ and K^+^ channel blockers raise important caveats for this analysis. First, these agents inhibit multiple subtypes of voltage-gated Ca^2+^ and K^+^ channels and are likely to block other classes of channels not addressed here that could impact glioma cell properties. Second, transcript levels do not necessarily correlate with levels of active proteins in cell membranes, which depend on translation, membrane localization, and post-translational modifications that dictate functional impact. 

No differences between the GBM subtypes for *AQP1*, *SCNN1A*, *KCNN2*, or *KCNN3* expression levels were apparent (Figure 8). The uniform expression of AQP1 is consistent with the relatively constant blocking effect of AqB013 when applied alone, and the ubiquitous additional block of invasion that was observed in combined treatments with AqB013. An exception to the pattern is apamin, which alone was less effective in U251-MG than U87-MG, perhaps consistent with a trend toward reduced levels of *KCNN3* expression in the classical subtype, or due to other effects noted as caveats above. Interestingly, amiloride combined with AqB013 yielded significant additive inhibitory effects on the invasion of U87-MG, a feat that was not replicated in U251-MG despite similar transcript levels of *SCNN1A* being observed between the proneural and classical subtypes (Figure 8). This may be due to a significantly higher expression of the amiloride-sensitive acid-sensing ion channel 1 (*ASIC1*) [85] in the proneural subtype by comparison to the classical subtype (https://gbm-biodp.nci.nih.gov last accessed on 28 December 2022). It has been shown previously in in vitro glioma models that ASIC1 colocalizes and works in tandem with ENaC subunits to facilitate glioma migration [86], an interaction that might have been disrupted upon treatment with amiloride and further decreased cellular invasion in proneural-like U87-MG cells compared to classical-like U251-MG cells. Apamin-sensitive Ca^2+^-activated K^+^ channels K_Ca_2.2 (*KCNN2*) and K_Ca_2.3 (*KCNN3*) showed high transcript levels in both the proneural and classical GBM subtypes. The widespread involvement of K_V_ channels with invasion-related events, such as focal adhesion assembly and turnover and ECM degradation [87,88,89], suggest complex responses to pharmacological treatments are likely, and ultimately analyses will need to rely on empirical testing, not extrapolation from transcriptomic analyses. 

XA acts as a positive allosteric ligand at metabotropic glutamate receptors mGluR2 and mGluR3 (encoded by *GRM2* and *GRM3*, respectively) [39] to influence Ca^2+^-dependent motility pathways [40,41]. No effects of XA were exerted on either U87-MG or U251-MG spheroid in the dispersal assays, which might reflect in part the similar levels for *GRM2* and *GRM3* transcripts between proneural and classical subtypes of GBM (Figure 8), and other factors. Not tested here is the possibility that assembly of GBM cells into aggregates could change the levels of expression of classes of channels and receptors that add complexity to the responses to pharmacological agents. Metabotropic glutamate receptors are enriched in U87-MG spheroids as compared to expression levels in the same cells cultured as a monolayer [90]. Differences in results between the transwell and the spheroid models of invasion could provide insights into aspects of cellular motility that are required for the detachment and dispersal of secondary nodes, which differ from the requirements for permeation of dissociated cells through transwell filters. 

Spheroids are 3D cell structures that have been widely used for oncological drug discovery studies. Similar to native tumors, spheroids form three distinct cell layers: an outer proliferating cell layer, a quiescent middle cell layer, and a necrotic core; and cells can disseminate and reattach to adherent surfaces in a manner reminiscent of metastasis [54,91]. Presumed to recapitulate the heterogeneous complexities of solid tumors to a greater extent than do cell suspensions, spheroids are thought to model in vivo tumor cell motility more closely than do transwell assays, though this model does have limitations. Particle dispersion from aggregate masses of cells depends on complex cellular machinery including anchorage proteins, cell-cell adhesion molecules, and metalloproteinases [15,16,17,18]. A three-dimensional spheroidal structure adopted by T98G glioblastoma cells was suggested to be critical for the induction of cancer stem cell-like cells, characterized by enhanced migration and decreased sensitivity to x-ray irradiation and temozolomide, presenting survival advantages not displayed with T98G cells cultured in monolayers [92]. In addition, it is possible that access of pharmacological agents wanes over the 96-hour time course of the assay; cells in spheroid cores have been found to promote an acidic microenvironment associated with decreased cellular uptake of drugs [93].

We noted that treating U87-MG and U251-MG spheroids with a combination of AqB013 and nifedipine, amiloride, apamin, 4-AP, or CNQX yielded no augmentation of the inhibition of particle dispersal. Treatment of U87-MG spheroids with XA or caelestine C did not impede particle dispersal as compared to the vehicle control. Reduced particle dispersal in co-treatments with AqB013 and XA did not show additive effects over levels seen with AqB103 alone. 

Compounds tested in this study were used at concentrations that exerted no significant cytotoxicity in U87-MG, U251-MG, or astrocytes, supporting the conclusion that the observed reductions in cellular invasion reflected anti-invasion actions rather than cell death. Normal astrocyte survival in the presence of these inhibitors might be an early indication that treatment with these inhibitors could be tolerated in the neuronal environment that surrounds glioblastoma tumors, providing an encouraging step toward the development of localized glioblastoma therapeutics that spare the neighboring brain tissue. The main limitations of the methods employed here included the promiscuity of nifedipine and 4-aminopyridine, potential disparities between the transcript levels and active protein expression of the channels of interest, and the inherent exclusion of cells within the inner layers of spheroid structures to drug exposure by comparison to the outer layer. Nevertheless, results here support the idea that inhibition of AMPA glutamate receptor channels in combination with inhibition of AQP1 appears to be an attractive means for limiting cellular motility in GBM tumors.

## 5. Conclusions

Membrane channels and receptors are emerging as key mechanisms in facilitating the high motility of glioblastoma tumor cells, tested here by pharmacological targeting of classes of aquaporins, voltage-gated ion channels, and ionotropic receptors shown to be upregulated in transcriptomic analyses of human GBM biopsies. Results showed antagonists of AQP1, Na^+^, K^+^, and Ca^2+^ channels; Ca^2+^-activated K^+^ channels; and AMPA/kainate-type glutamate receptors, and an agonist of metabotropic glutamate receptors, are promising targets for the optimization of small-molecule inhibitors of glioblastoma tumor motility. Anti-invasive properties of xanthurenic acid and caelestine C were demonstrated in transwell invasion assays. Concurrent inhibition of AQP1 augmented the block of invasion in glioblastoma, indicating channels could serve complementary roles in facilitating motility. Combinations of inhibitors which yielded the most potent block of invasion differed between U87-MG and U251-MG, suggesting further potential to tailor treatments to specific GBM subtypes. Understanding the molecular and signaling mechanisms that drive glioblastoma survival and invasive spread will be essential for the development of targeted therapies, ideally framing novel combination regimens that can improve quality of life for affected populations of patients.

## Figures and Tables

**Figure 1 cancers-15-01027-f001:**
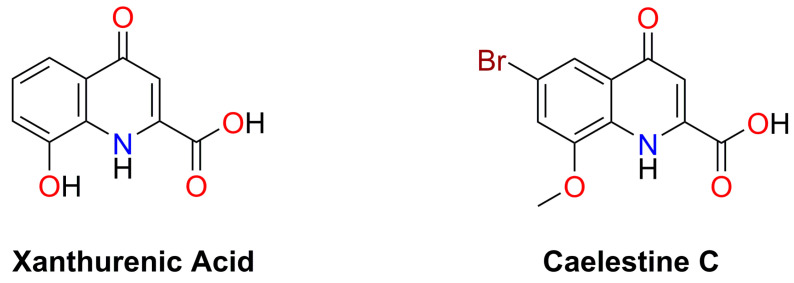
Chemical structures of the natural compounds xanthurenic acid and caelestine C sourced from the Davis Open Access Natural Product-based Library that show potent activity in reducing GBM motility at non-toxic micromolar concentrations.

**Figure 2 cancers-15-01027-f002:**
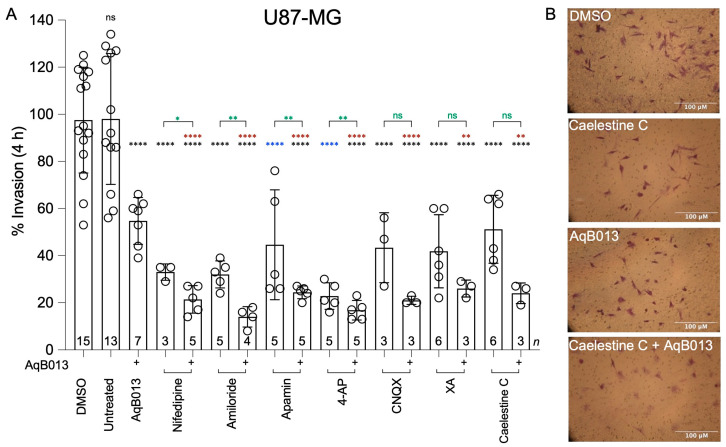
Effects of pharmacological channel inhibitors and natural products on U87-MG invasion. (**A**) Compiled data in bar histograms show invasion standardized as a percentage to the mean values of vehicle or untreated controls. Statistical significance was determined with one-way ANOVA and Dunnett’s post-hoc tests. Statistically significant differences for individual agents with (+) and without AqB013 are shown in green. Differences between combined treatments and AqB013 alone are indicated in red. Differences between treatment groups and corresponding controls are indicated in black (vehicle) or blue (untreated). Doses of compounds were: AqB013 (14 μM); nifedipine (25 µM); amiloride (10 µM); apamin (10 µM); 4-AP (250 µM); CNQX (30 µM), XA (0.3 µM); caelestine C (0.1 µM). (**B**) Illustrative images are shown for stained U87-MG cells that traversed ECM-layered transwell filters by 4 h; in control (DMSO), caelestine C, AqB013, or combined AqB013 + caelestine C treatments (doses are the same as in (**A**)). *p* values less than 0.05 are summarized with one asterisk. *p* values less than 0.01 are summarized with two asterisks. *p* values less than 0.0001 are summarized with four asterisks.

**Figure 3 cancers-15-01027-f003:**
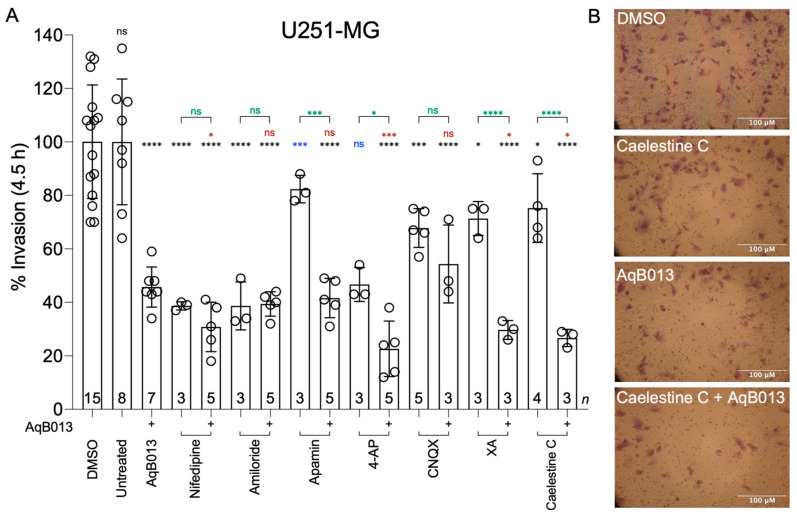
Effects of pharmacological channel inhibitors and natural compounds on U251-MG invasion. (**A**) Compiled data shown in bar histograms show invasion at 4.5 h, standardized to the mean control (vehicle or untreated). Statistically significant differences were determined using one-way ANOVA with Dunnett’s post-hoc tests; comparisons between treatments and matched controls are shown for vehicle (black) and untreated (blue). Differences for compounds with and without AqB013 are shown in green. Differences between combined treatments and AqB013 alone are in red. Doses of compounds were as specified in Figure 2. (**B**) Illustrative images of stained U251-MG cells that traversed the ECM-barrier transwell filters are shown at 4.5 h in treatments with DMSO, caelestine C, AqB013, or caelestine C + AqB013 combined. *p* values less than 0.05 are summarized with one asterisk. *p* values less than 0.001 are summarized with three asterisks, and *p* values less than 0.0001 are summarized with four asterisks.

**Figure 4 cancers-15-01027-f004:**
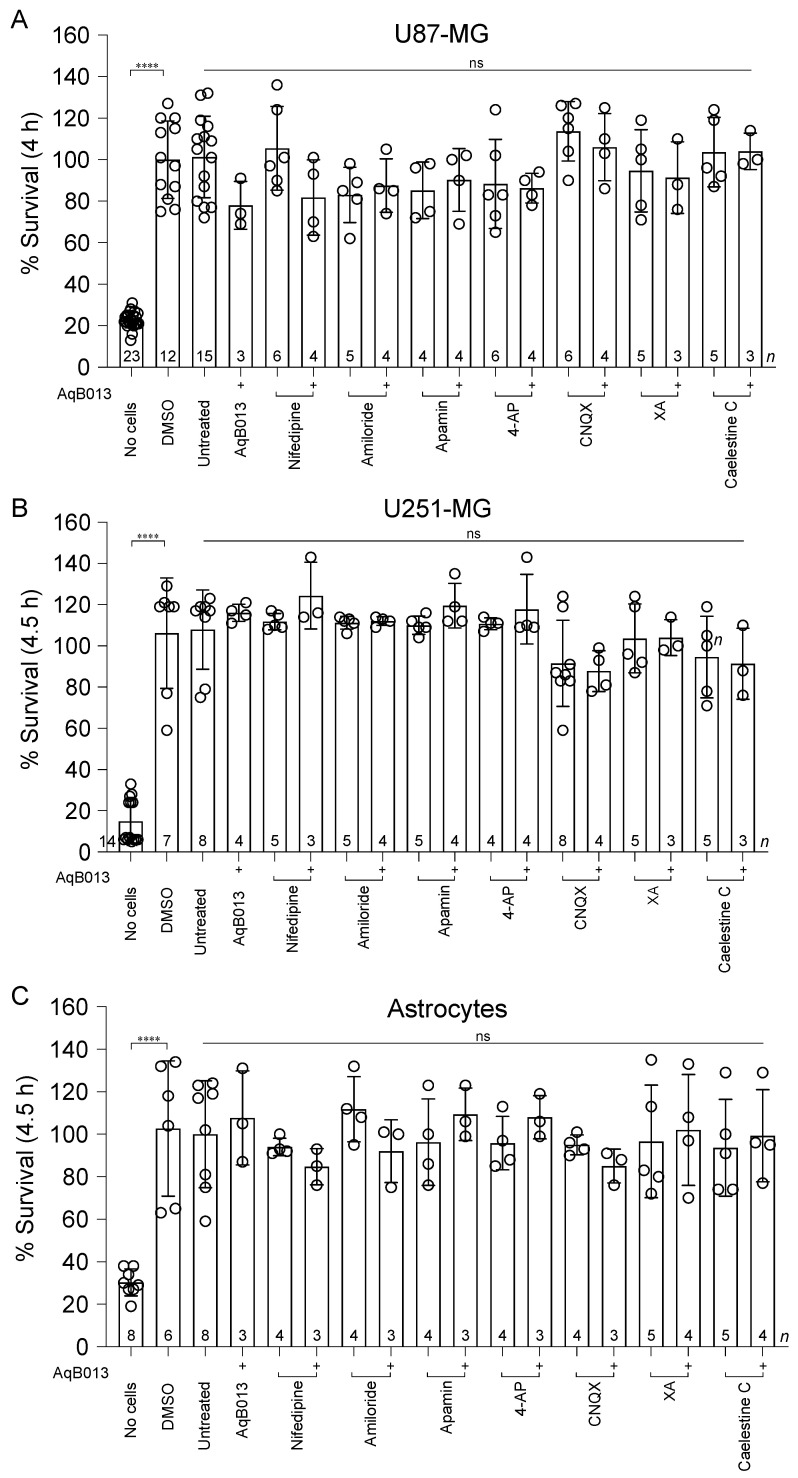
Lack of treatment effects on cell viability of GBM and 1321N1 astrocyte lines. Compiled data for U87-MG (**A**), U251-MG (**B**), and 1321N1 astrocytes (**C**) are shown in bar histograms. Viability data (mitochondrial metabolic activity) were standardized as a percentage (%) of the mean value for vehicle-treated control cells, at 4 h (U87-MG) or 4.5 h (U251-MG and 1321N1) post-treatment. Compounds were applied at: AqB013 (14 μM); nifedipine (25 µM); amiloride (10 µM); apamin (10 µM); 4-AP (250 µM); CNQX (30 µM), XA (0.3 µM); caelestine C (0.1 µM). One-way ANOVA with Dunnett’s post-hoc tests showed no significant differences between treatment groups and vehicle controls. *p* values less than 0.0001 are summarized with four asterisks.

**Figure 5 cancers-15-01027-f005:**
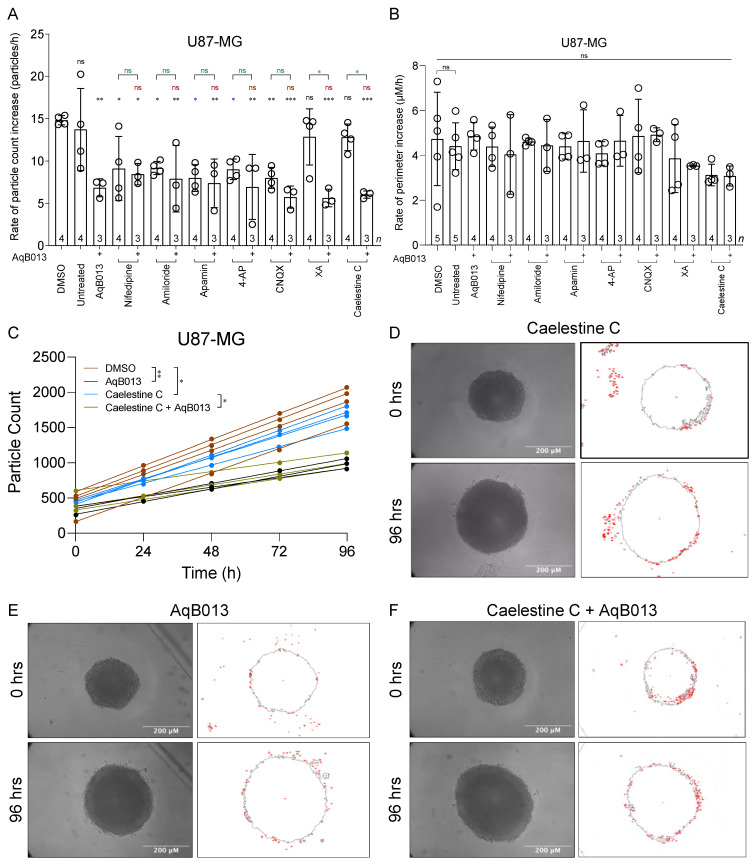
Inhibition of particle dispersal from U87-MG spheroid masses embedded in extracellular matrix, without loss of cell viability. (**A**) Bar histogram showing mean rates of particle dissemination (mean ± SD) from individual U87-MG spheroids in different treatments. Statistically significant results are represented by asterisks as *** *p* < 0.001, ** *p* < 0.01, * *p* < 0.05, and ns (not significant). (**B**) Growth rates (mean ± SD) measured as increased perimeter over time in different treatments. Statistically significant differences from control were assessed with one-way ANOVA and Dunnett’s post-hoc tests. (**C**) Total numbers of particles dispersed from individual spheroids as a function of time are shown for 0.2% DMSO (brown), 28 μM AqB013 (black), 0.2 µM caelestine C (blue), and combined application of 0.2 µM caelestine C and 28 μM AqB013 (khaki); *n* = 4 each. Fitted slope values provided the rates compiled in (**A**). Statistical significance was determined by repeated measures ANOVA with the Geisser-Greenhouse correction. (**D**–**F**) U87-MG spheroid perimeters measured at 0 and 96 h during treatments with (**D**) caelestine C (0.2 µM), (E) AqB013 (28 μM), or (**F**) the same doses of caelestine C and AqB013 in combination. No reductions in spheroid growth rates (perimeter lengths, shown as grey outline) were seen; in contrast decreased numbers of secondary nodes (red) were evident with AqB013.

**Figure 6 cancers-15-01027-f006:**
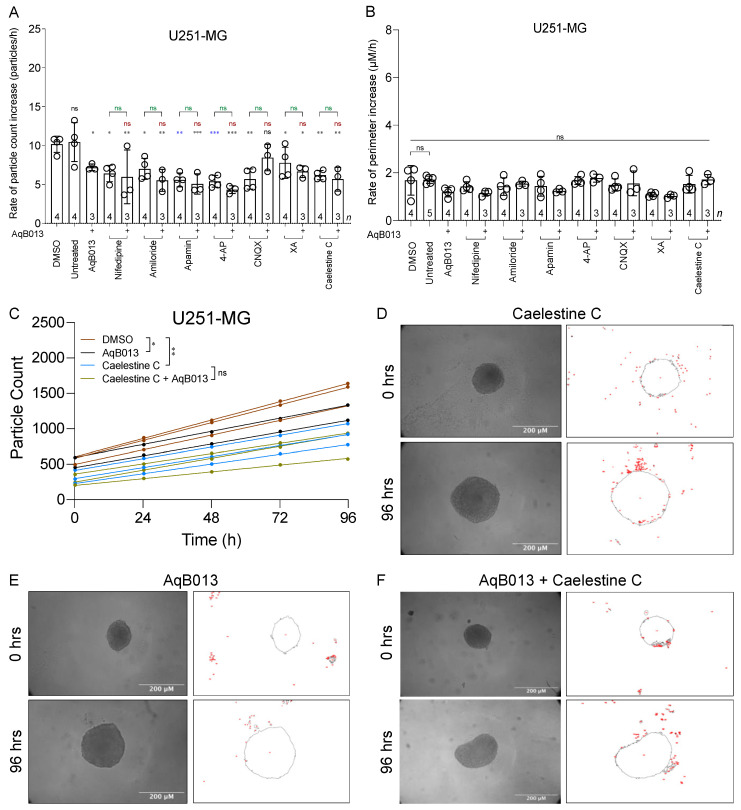
Inhibition of particle dispersal from U251-MG spheroid masses embedded in extracellular matrix, without loss of cell viability. (**A**) Effects of pharmacological agents on rates of particle dissemination from individual U251-MG spheroids (mean ± SD). Statistically significant results are represented by asterisks as *** *p* < 0.001, ** *p* < 0.01, * *p* < 0.05, and ns (not significant). (**B**) Rates of perimeter increases for individual U251-MG spheroids (mean ± SD) indicating growth in all treatments. Statistically significant differences from control were determined by one-way ANOVA with Dunnett’s post-hoc tests. (**C**) Total numbers of particles dispersed from individual U251-MG spheroids treated with 0.2 µM caelestine C and 28 μM AqB013 in combination (khaki), or AqB013 (black) or caelestine C (blue) alone at the same doses, as compared to DMSO control (brown), fit by linear regression. Statistical analyses used repeated measures ANOVA with the Geisser-Greenhouse correction. (**D**–**F**) Images of U251-MG spheroids at 0 and 96 h of treatments with (**D**) caelestine C (0.2 µM), (**E**) AqB013 (28 μM), or (F) both caelestine C + AqB013 (perimeters shown as grey outlines).

**Figure 7 cancers-15-01027-f007:**
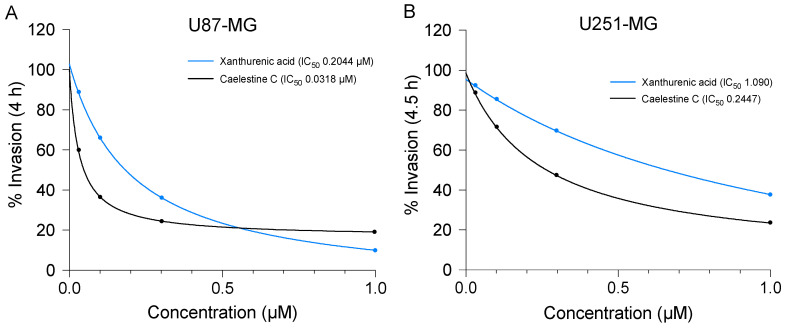
Potent inhibition of transwell invasion in glioblastoma cell lines by low-dose treatments with compounds linked glutamatergic receptor modulation. Dose-response curves show concentration-dependent inhibition of invasion for U87-MG (**A**) and U251-MG (**B**) in treatments with XA and caelestine C. Half-maximal inhibitory concentrations (IC_50_) were estimated by non-linear regression fits using GraphPad Prism software; values are listed in figure keys.

**Figure 8 cancers-15-01027-f008:**
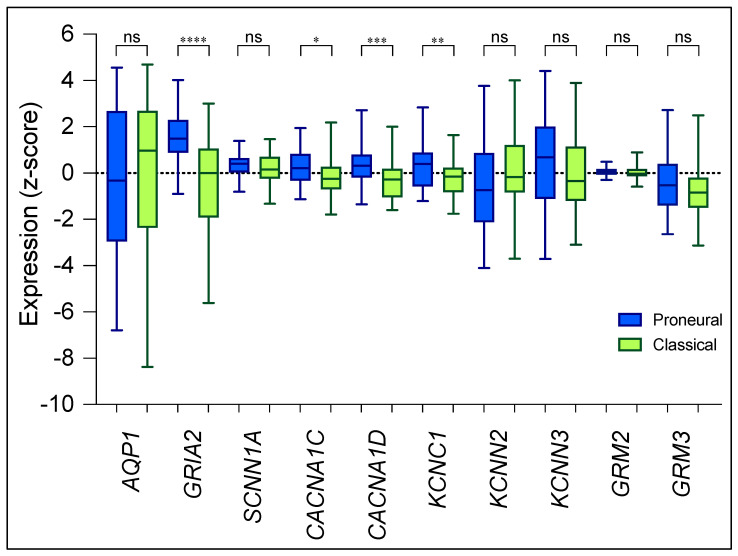
Transcript levels for channel targets of interest measured by RNAseq in human biopsy samples of proneural and classical glioblastoma subtypes. Transcript levels for aquaporin-1 (AQP1), AMPA glutamate receptor GluR2 (GRIA2), and selected ion channels (Epithelial Na^+^ channel (ENaC), SCNN1A; Ca_V_1.2, CACNA1C; Ca_V_1.3, CACNA1D; K_V_3.1, KCNC1; K_Ca_2.3, KCNN3; K_Ca_2.2, KCNN2; mGluR2, GRM2; and mGluR3, GRM3), were compiled from the GBM Bio Discovery Portal database. Z-scores for transcript levels in each patient reflect distance from the mean transcript level for the full population (*n* = 197) measured in standard deviations. Significant differences in transcript levels for each gene when comparing proneural (blue, *n* = 56) with classical (green, *n* = 53) subtypes were measured using unpaired t-tests with Welch’s correction. Statistically significant results are represented by asterisks as **** *p* < 0.0001, *** *p* < 0.001, ** *p* < 0.01, * *p* < 0.05, and ns (not significant).

## Data Availability

Data will be made available on request to the corresponding author.

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
