# Peer review of "Pharmacological Inhibition of Membrane Signaling Mechanisms Reduces the Invasiveness of U87-MG and U251-MG Glioblastoma Cells In Vitro"

_cancers, 2023, doi:10.3390/cancers15041027_

Round 1

Reviewer 1 Report

The manuscript submitted is well organized and seems to be good according to used methods and obtained results. 

I have some suggested comments : 

- In the title: avoid the use of . after the title 

- Please reduce the number of key words 

- Figures 2, 3, 4, and 5 should be improved 

- The introduction should be improved by incorporating recent data about natural bioactive compounds as anticancer drugs 

- The first paragraph of the discussion part should be improved by introducing this part by a paragraph concerning the main objective of this investigation.

Author Response

In the title: avoid the use of . after the title

     This full stop has been removed

Please reduce the number of key words

     Done as requested

Figures 2, 3, 4, and 5 should be improved

     Images have been reformatted to improve the resolution to the maximum possible.

The introduction should be improved by incorporating recent data about natural bioactive compounds as anticancer drugs

     Recent findings about bioactive compounds chlorotoxin, apamin and melittin as anticancer drugs have now been included in the introduction. (lines 115-118 in the highlighted pdf)

The first paragraph of the discussion part should be improved by introducing this part by a paragraph concerning the main objective of this investigation.

     An introductory paragraph has been added to the Discussion section to introduce the objectives of the investigation. (lines 119-130).

Reviewer 2 Report

The manuscript entitled” Pharmacological inhibition of membrane signaling mechanisms reduces the invasiveness of U87-MG and U251-MG glioblastoma cells in vitro” by Varricchio A. and collaborators focused on reduction glioblastoma invasive behavior using a wide range of pharmacological inhibitors. The subject is of high interest. Some concerns are listed below:

Both the Abstract and Introduction sections are too long. Moreover, the authors summarize their results in the Introduction section (lines 118-119, 131-135). Please check and adjust both sections.

Please add information regarding the selection of pharmacological inhibitors concentrations.

Please enhance the quality of all images.

Lines 300-307, please remove or relocate. That information is more suitable for Discussion section.

The images for spheroid spreading assay (Figure5 D-F and Figure6 D-F) are provided only for glioblastoma cells treated with Caelestine C, AqB013 and Caelestine C + AqB013. Please add the images for spheroid spreading assay corresponding to the other used inhibitors as Supplementary files.

Lines 371-385, should be relocated after the results of viability assay.

The limitations of the study, which are mentioned throughout the Discussion section should be presented in a separate section.

Quantification of some specific markers associated with invasive behavior would certainly strengthen the authors findings.     

Author Response

The Introduction sections is too long. Moreover, the authors summarize their results in the Introduction section (lines 118-119, 131-135). Please check and adjust both sections.

     The Introduction section has been made more concise following the deletion of two paragraphs of text. The summaries of the results have been shortened and a summary statement  was relocated in the introductory paragraph of the Discussion section (lines 395-396).

Please add information regarding the selection of pharmacological inhibitors concentrations.

     Information pertaining to the selection of the concentrations of pharmacological inhibitors tested has been added to the Materials and Methods section under 2.2 Pharmacological agents. (lines 165-170)

Please enhance the quality of all images.

     Images have been reformatted to improve the resolution to the maximum possible for the imaging system.

Lines 300-307, please remove or relocate. That information is more suitable for Discussion section.

     This has been relocated to the Discussion section. (lines 566-574)

The images for spheroid spreading assay (Figure5 D-F and Figure6 D-F) are provided only for glioblastoma cells treated with Caelestine C, AqB013 and Caelestine C + AqB013. Please add the images for spheroid spreading assay corresponding to the other used inhibitors as Supplementary files.

     Images have now been provided to cover all treatments in both cell lines as Supplementary Figure 2 (U87-MG) and Supplementary Figure 3 (U251-MG). Image quality has been enhanced as much as possible. (lines 358-360, and new Supp Figs 2 and 3)

Lines 371-385, should be relocated after the results of viability assay.

     A statement was added as suggested noting that in the spheroid model, spheroid perimeter was used as a metric to determine the effect of each of the drug treatments on spheroid growth. (Iines 311-312)

The limitations of the study, which are mentioned throughout the Discussion section should be presented in a separate section.

   Noting the limitations of each group of experiments in context throughout the Discussion section will be helpful for some readers and has been retained. However, to address the concern raised, a full overview of the limitations also has been added at the end of the discussion, just before the significance section. (lines 581-591)

Quantification of some specific markers associated with invasive behavior would certainly strengthen the authors findings.

    A new Supplementary Figure 1 was added. These results summarize quantitative data measuring the elongation index of transwell invaded cells, to provide a tool for statistical analyses of the changes in morphology that cells undergo when adopting an elongated spindle shape characteristic of motile cells. (lines 285-291; and new Supp Fig 1)

Round 2
